# Challenges for funders in monitoring compliance with policies on clinical trials registration and reporting: analysis of funding and registry data in the UK

Rachel L Knowles ,[1] Kam Pou Ha,[1] Julia Mueller,[2] Frances Rawle,[1] Rosa Parker[1]

[1]Medical Research Council, London, UK
[2]Division of Population Health, Health Services Research and Primary Care, University of Manchester, Manchester, UK

**Correspondence to**
Dr Rachel L Knowles;
rachel.knowles@mrc.ukri.org

## ABSTRACT

**Objectives** To evaluate compliance by researchers with funder requirements on clinical trial transparency, including identifying key areas for improvement; to assess the completeness, accuracy and suitability for annual compliance monitoring of the data routinely collected by a research funding body.

**Design** Descriptive analysis of clinical trials funded between February 2011 and January 2017 against funder policy requirements.

**Setting** Public medical research funding body in the UK.

**Data sources** Relevant clinical trials were identified from grant application details, post-award grant monitoring systems and the International Standard Randomised Controlled Trial Number (ISRCTN) registry.

**Main outcome measure** The proportion of all Medical Research Council (MRC)-funded clinical trials that were (a) registered in a clinical trial registry and (b) publicly reported summary results within 2 years of completion.

**Results** There were 175 grants awarded that included a clinical trial and all trials were registered in a public trials registry. Of 62 trials completed for over 24 months, 42 (68%) had publicly reported the main findings by 24 months after trial completion; 18 of these achieved this within 12 months of completion. 11 (18%) trials took >24 months to report and 9 (15%) completed trials had not yet reported findings. Five datasets were shared with other researchers.

**Conclusions** Compliance with the funder policy requirements on trial registration was excellent. Reporting of the main findings was achieved for most trials within 24 months of completion; however, the number of unreported trials remains a concern and should be a focus for future funder policy initiatives. Identifying trials from grant management and grant monitoring systems was challenging therefore funders should ensure investigators reliably provide trial registries with information and regularly update entries with details of trial publications and protocols.

## INTRODUCTION

It is estimated that around 1 million clinical trials have been carried out since 1948 when the UK Medical Research Council

**Strengths and limitations of this study**

► The study evaluated the contribution made by different data sources to constructing a comprehensive dataset that allowed an audit of grant holders' compliance with one funder's clinical trial transparency policy.

► Information in the funder's databases for each trial was compared with the public registry entry for accuracy.

► Using grant applications, annual grant monitoring and trial registry data, the proportion of all funded clinical trials that were (i) publicly registered and (ii) for which results were publicly reported was calculated for a 6-year period.

► Study strengths included the completeness of the dataset and careful validation of registration and publication details.

► Analyses were limited by the small number of trials per year.

(MRC) conducted a landmark randomised controlled trial on streptomycin in pulmonary tuberculosis.[1] Although responsibility for funding phase III clinical trials in the UK transferred to the National Institute for Health Research (NIHR) in 2006, the MRC, as part of UK Research and Innovation, continues to directly fund early phase clinical trials, phase III global health trials, other clinical and public health intervention studies, and to provide underpinning funding for trials methodology research and clinical trials units.

Serious concerns have been raised about the failure by researchers to report findings from a high proportion of clinical trials, as well as the selective reporting of 'positive' findings.[2–6] This represents not only a loss of valuable research findings but may lead to potential overestimation of the benefits of new treatments, with the risk that ineffective

**BMJ**

or harmful drugs or interventions are implemented in clinical practice.[3 7] Public trials registers, which facilitate the prospective registration of clinical trials, play a key role in reducing selective outcome reporting and publication bias[8] and the International Committee of Medical Journal Editors (ICMJE) requires prospective registration of trials as a condition of publication in its member journals.[9] The WHO has established an International Clinical Trials Registry Platform (ICTRP), which collects information on ongoing and completed trials from a network of primary registries across the world and provides a single point of public access.[10 11]

The AllTrials[12] campaign was established to actively promote clinical trials transparency '*All trials registered All trials reported*', and funder signatories to the AllTrials petition, such as the UK MRC, subsequently strengthened requirements for grant recipients to prospectively register clinical trials and publicly disseminate results (online supplementary file 1). Despite this, a UK House of Commons Science and Technology Committee (STC) Inquiry into Research Integrity: Clinical Trials Transparency in 2018 concluded that '*registration is not yet universal … and reported outcomes do not always align with the original study proposal*'. The Committee strongly recommended that compliance with reporting requirements should be monitored by funders and regulators and published at an individual trial level.[13] This call for improved monitoring of compliance with registration and reporting policies echoed the WHO *Joint statement on public disclosure of results from clinical trials* to which 21 funding agencies have now signed up.[14]

As a founder signatory to the WHO Joint statement, the MRC committed to developing ongoing monitoring of compliance with its *Policy on Open Research Data from Clinical Trials and Public Health Intervention Studies* published in October 2016[15] (box 1). An initial review of research grants awarded and active over the 5-year period from 01 February 2011 to 31 January 2016 identified 107 clinical trials of which 101 (94%) were registered in a public trials registry. From 40 completed trials, 33 (82%) reported at least one publication.[16]

In this paper, we present a subsequent evaluation undertaken in 2018 that included MRC funding for awards involving a clinical trial up to 31 January 2017. Our aims were (1) to evaluate compliance by Principal Investigators (PIs; lead researcher or Grant Holder) with MRC's clinical transparency requirements as set out in the *Policy on Open Research Data from Clinical Trials and Public Health Intervention Studies* published in 2016[15] and to identify key areas for improvement, and (2) to assess data completeness and accuracy as a basis for the further development of annual compliance monitoring.

## METHODS
### Data sources
The initial step was to compile a single monitoring dataset that included all awards made by the MRC during

---

> **Box 1    Summary of MRC policy requirements (October 2016)\***
>
> **Registration**
> 1. Prospective registration of every MRC-funded clinical trial in the ISRCTN registry\* prior to recruitment of the first participant.
> (\**Other WHO primary registries accepted if a prior agreement with the MRC*)
> 2. Provision of the registry number to the MRC within 12 months of registration.
> 3. Reviewing (and updating if appropriate) the registry entry at least once per year until the trial has reported main results.
>
> **Reporting**
> 4. Public report of the trial protocol—there should be a link to this from the registry entry (the protocol may be added as a supplementary document in the registry entry).
> 5. Public report of the trial's main results in a timely manner (usually within 12 months); there should be a link to the open accessible report/publication from the registry entry (and/or a report may be added as a supplementary document in the registry entry).
>
> **Data Sharing**
> 1. Preparation of the trial dataset for sharing with external researchers is encouraged; researchers are expected to provide details of any datasets/databases created as part of an award in their annual report to the MRC.
>
> \*This table summarises requirements from the MRC policy.[15]
> ISRCTN, International Standard Randomised Controlled Trial Number; MRC, Medical Research Council.

the audit period from 01 February 2011 until 31 January 2017, which included a clinical trial meeting the WHO definition: '*any research study that prospectively assigns human participants or groups of humans to one or more health-related interventions to evaluate the effects on health outcomes*'.[17] The dataset included research grants, fellowship awards, global health trials and any developmental clinical study award in which preclinical work was followed by a clinical trial. Trials funded through MRC Research Unit and MRC/University Unit programmes to clinical trials units or joint-funding not directly managed by the MRC, for example, the NIHR Efficiency and Mechanism Evaluation (EME) programme, were excluded. Information was extracted from four sources (table 1): the MRC grants application and award database; the annual output, outcome and reporting system (Researchfish); a listing of joint-funded global health trials for which the MRC had oversight; and an external dataset provided by the International Standard Randomised Controlled Trial Number (ISRCTN) registry.[18]

### Search methods (2017)
Basic keyword searches (online supplementary file 2) of MRC grants application and annual monitoring databases for the 2017 audit yielded 373 funded awards that potentially included a trial (figure 1), and for 112 of these a trial registry number had been reported to the MRC. To complete missing entries, the grant reference, study title, acronym and PI name were entered into the three clinical

**Table 1** Sources of data for monitoring

| Database | Purpose of databasef | Data entered by | Data collected (relevant to monitoring) | Search strategy |
|---|---|---|---|---|
| MRC Grants Application and Award database | MRC application management and monitoring Data re publicly available in UKRI Gateway to Research[22] | Applicants | Data entered as part of the funding application and award process. 1. Award start and end dates 2. PI contact details 3. Study title and summary 4. Amount of award | Automated keyword search in title, abstract, study summary |
| MRC annual monitoring of awards (Researchfish) | MRC monitoring of activity and outputs from research funding. Data collected via the Researchfish platform Data are publicly available in UKRI Gateway to Research[22] | Named PI receiving funding | Annual report/update to the MRC on 'live' or recently ended awards every February. Includes: 1. Clinical trial 'flag', ie, trial associated with an award 2. Trials registry number 3. Publications and other outputs 4. Datasets created/shared | Automated keyword search in title, abstract, study summary, outputs, publication fields Automated search for clinical trials 'flag', entry for registry number |
| MRC governance monitoring of global health trials | MRC governance of global health clinical trials Data not publicly available | MRC staff | Information entered for governance and oversight 1. Record of Trial Steering Committee dates 2. Changes to trial start and end dates | Manual search of study summary or notes to confirm study design and identify trials |
| ISRCTN registry[18] | Publicly accessible and searchable database of individual clinical trials Trial information is regularly submitted to WHO ICTRP portal where it is publicly available[51] | Research team | Data entered at registration then updated or amended by the research team as required. 1. Trial start and end dates 2. Contact person 3. Registration date 4. Links to protocol and main results 5. Trial funder 6. Trial sponsor 7. Date entry last updated | Automated search for Medical Research Council or MRC in funder and sponsor fields |

ICTRP, International Clinical Trials Registry Platform; ISRCTN, International Standard Randomised Controlled Trial Number; MRC, Medical Research Council; PI, Principal Investigator; UKRI, UK Research and Innovation.

trials registries most commonly used for UK research, the ISRCTN registry,[18] ClinicalTrials.gov[19] and European Union Clinical Trials Registry (EUCTR),[20] followed by an online web search using these terms.

Registered (n=124) and unregistered studies that did not meet the WHO clinical trial definition were excluded, as were registered trials (n=67) which began before the audit period. PIs of 25 unregistered but potential trials were contacted for further information; registry numbers were provided for 9 trials and 15 were reported to be preclinical. One trial was unregistered but the research team proceeded to register it retrospectively. In total, 117 trials were included in a preliminary review in 2017.

Additional sources and updated searches (2018)
Following the collection of annual researcher reports in February 2018, the MRC 2017 dataset was updated and 16 studies were excluded (figure 1). Eleven new awards made between 01 February 2016 and 31 January 2017 were added, as well as five studies that had progressed from preclinical research to a clinical trial during this period. Global health trials undergo additional governance monitoring by the MRC and this list yielded 28 additional trials not previously included.

A search of the independent ISRCTN registry dataset yielded 132 entries for trials starting on or after 01 February 2011 with the MRC named as the funder or

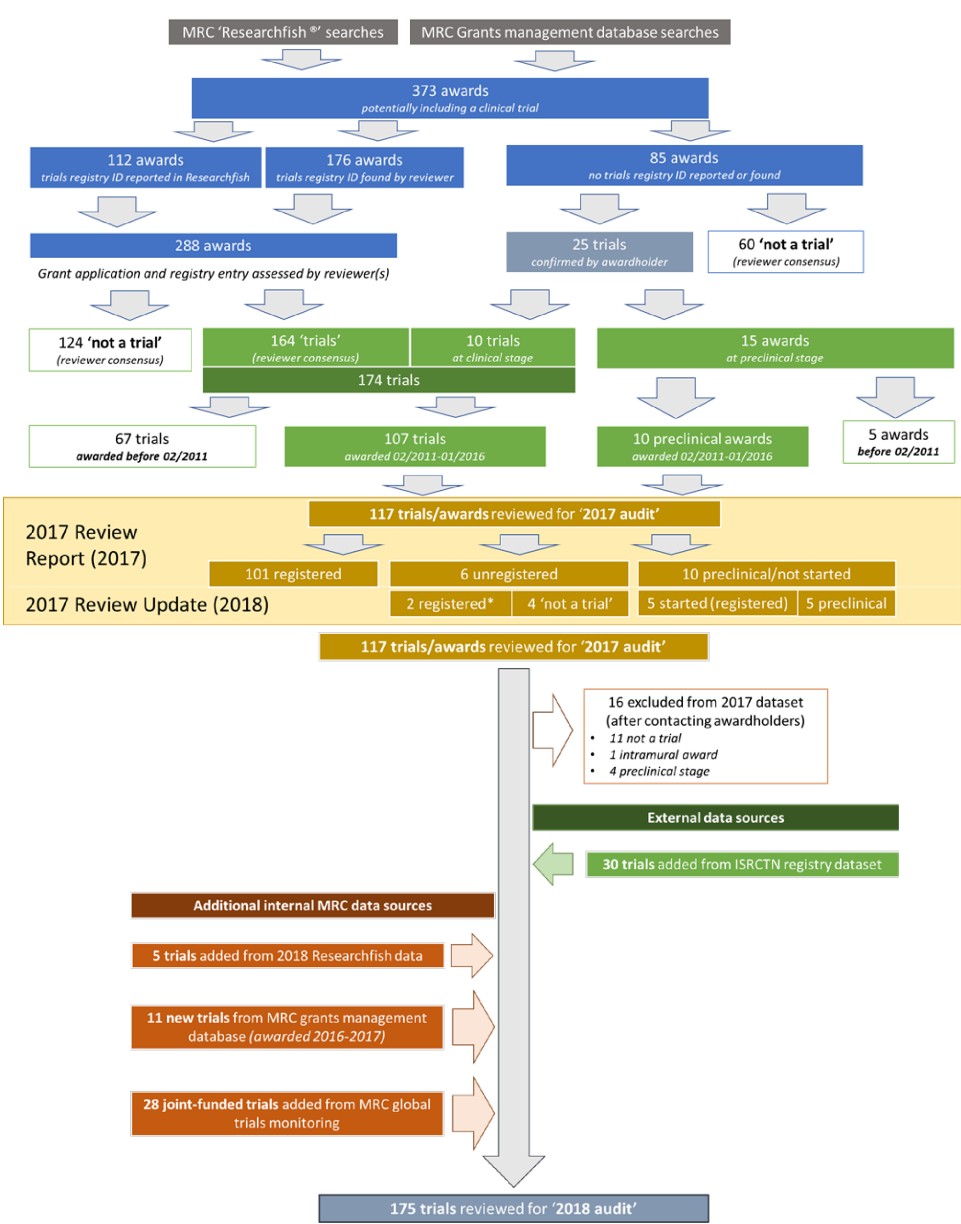

**Figure 1** Flow diagramm of trials identified for the review. ISRCTN, International Standard Randomised Controlled Trial Number; MRC, Medical Research Council.

sponsor. After review, 90 trials were not eligible (funder/sponsor was incorrect), 12 trials were already included and 30 previously unidentified trials were added to the review dataset.

### Confirming eligibility
A total of 175 awards made by the MRC between 01 February 2011 and 31 January 2017 were identified as including a clinical trial and eligible for inclusion in the 2018 review dataset.

Prior to commencing the review, project titles and abstracts of all awards were reviewed to confirm that they included a clinical trial. They were classified as a *trial, not a trial, trial not started* (preclinical) by two reviewers. PIs were contacted if uncertainty remained about whether the award included a clinical trial, or if registry numbers or main results publications could not be identified.

### Evaluating the monitoring process: data completeness and accuracy
Each source contributing to the review dataset was individually assessed to determine the proportion of the total dataset that was ascertained.

To assess the accuracy of the review dataset, specific data items were compared between the MRC award information and the registry entry for each trial; this included checking MRC grant reference numbers, award/trial dates, PI name and research organisation, and publications. Information from this comparison, as well as reviewing how search parameters such as funder names were recorded, was explored for trials that were and were not identified by each source to determine the reasons why differences occurred.

### Evaluating researcher compliance with policy requirements

Analyses of compliance with the MRC *Policy on Open Research Data from Clinical Trials and Public Health Intervention Studies* (published October 2016) included the number and percentage of:

1. *All awards* that were registered in any public trials registry and the percentage prospectively registered in the ISRCTN (as required by the policy).
2. *All registry entries* that had a link to the study protocol or publication/report.
3. *All completed awards* that had publicly reported main, or summary, results.
4. *All awards* that reported creating a dataset for sharing beyond the study team/collaborators.

Changes in registration and reporting over time were conducted to investigate the impact of the MRC implementing the new policy that included *prospective* registration in 2016.[15] However, the audit period also covered the earlier introduction into award terms and conditions, in 2013, of the requirement for trials to be registered, either retrospectively or prospectively.[21] Evaluation of the timing of registration relative to trial start was restricted to the ISRCTN registry as this was the only registry that reported whether registration was prospective or retrospective.

As the information on award outputs is submitted annually by PIs using the Researchfish system, there can be a delay in reporting of registration or publications of up to 12 months. Therefore, analysis of registration was restricted to awards in which the trial had started at least 12 months prior to the analysis data extract being taken (01 February 2018) and analysis of publications was restricted to studies in which the funding award or trial end date (the later of these dates) was at least 12 months before the data extract (referred to as *completed awards*).

Research publications or reports were only considered to report the main results from the trial if they (i) were defined in the ISRTCN registry entry as including the main results, (ii) included information in the publication stating that they reported the main results, or (iii) were confirmed by the PI to report the main results. The funding *award* end date was considered to be equivalent to the trial end date; however, if the record in the registry stated that the *trial* end date was *after* the funding award end date, trials were considered completed on the *trial* end date. We refer throughout the paper to 'trial/award end' to capture this combined end point. Time to publication was therefore calculated using the *award* end date for most trials but using the *trial* end date reported in the registry when this was later than the award end date.

### RESULTS

#### Completeness of trials data from different sources

The contribution that each of the four data sources (table 1) made to the final review dataset was assessed. The original search strategy (online supplementary file 2) applied keywords to specific fields in the MRC grants application and annual monitoring databases but identified only 117 (67%) of 175 awards. Revised keyword searches (online supplementary file 2) were developed. These identified 913 studies, which were categorised by likelihood of including a clinical trial: category 1=highly likely to include a clinical trial (n=332); category 2=a clinical intervention study or trial (n=274); category 3=patient or cohort study (n=252); category 4=clinical trial methods (n=32); category 5=intervention study (eg, community interventions, economic evaluations of interventions and other non-randomised interventions; n=23). The revised searches had higher sensitivity than the original search, identifying 149 (85% of 175) awards in the final review dataset; however, they had poor positive predictive value as only 20% of retrieved awards included a clinical trial.

Assessment of annual reports to the MRC found that the trials registry number was not consistently reported and only 54 (29% of 175) awards were identified as including a trial via these data.

The 28 trials identified from global trials governance monitoring were not identified by the original keyword searches of MRC databases. They were not identified through the ISRCTN registry search, mainly because they received partnership funding and the MRC was not managing the grant.

The ISRCTN registry yielded 30 (17% of 175) trials that would otherwise have been missed. However, a further 90 (68% of the 132) studies identified by the ISRCTN search were excluded as checks showed that the study was either not a clinical trial or the trial did not receive MRC funding (although members of the study team may have received MRC funding for another research activity).

Therefore, both the search strategy and the accuracy of the registry entries presented significant challenges for accurate identification of relevant trials.

#### Accuracy of publicly available trials' data

Review of study abstracts, or full applications, enabled exclusion of most studies that did not include a clinical trial. However, it was necessary to contact PIs about 54 studies; 16 of these were subsequently excluded (figure 1).

Information on MRC awards, including award reference, PI, research organisation and award dates is published in UK Research and Innovation (UKRI) Gateway to Research.[22] Trials registry entries report trial dates, the PI's name, a contact person (who may also be the PI) and the organisation acting as the trial sponsor. In 175 registry entries, the following information from UKRI Gateway to Research was reported: the research organisation name (n=159 of 175; 91%), PI name (n=114; 65%), MRC award reference (n=60; 34%) and the MRC named as funder (n=67; 38%).

The dates of the trial in the registry entry matched the MRC award dates exactly in only five trials as most awards started before trial recruitment. The difference between these dates was even greater where awards included preclinical studies prior to the commencement of the clinical trial.

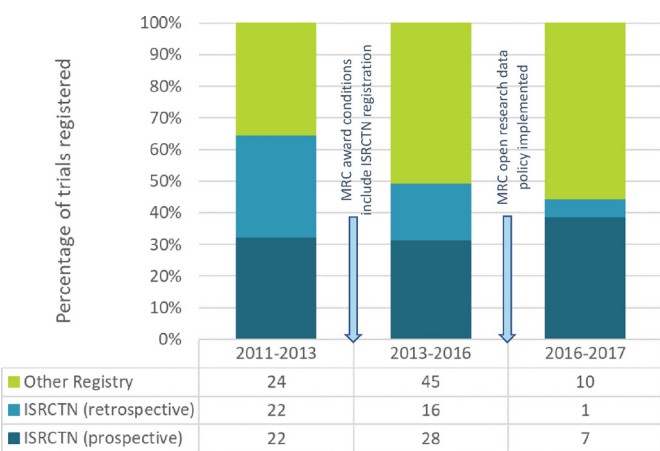

**Figure 2** Percentage of trials registered in ISRCTN or other registry (total n=175 trials). All trials are counted once only. Trials registered in ISRCTN are counted as ISRCTN only; trials registered in any other registry (and not in ISRCTN) are counted once in 'Other' registries. ISRCTN, International Standard Randomised Controlled Trial Number; MRC, Medical Research Council.

Of 3420 journal articles reported by researchers in Researchfish as outputs from the 175 MRC awards, only 6 were listed as the 'main results' publications in the ISRCTN registry.

### Trials registration

All 175 (100%) trials were registered in a public trials registry; 96 (55%) of these in the ISRCTN registry and 79 (45%) with other registries, such as ClinicalTrials.gov and EUCTR. One-quarter (n=43) were registered in more than one registry. Registry entries for 84 (76%) of the 111 trials ongoing in March 2019 had been updated since 01 January 2017. The trial protocol, or an electronic link to the protocol, was provided in only 63 (36% of 175) registry entries.

The percentage of MRC trials registered prospectively in the ISRCTN registry increased over time: in 2011–2013, this was 22 (50.0%) of 44, in 2014–2015, it was 28 (63.6%) of 44 and in 2016–2017, it was 7 (87.5%) of 8 trials (figure 2). In contrast, the introduction to award terms and conditions in 2013 of the requirement to use

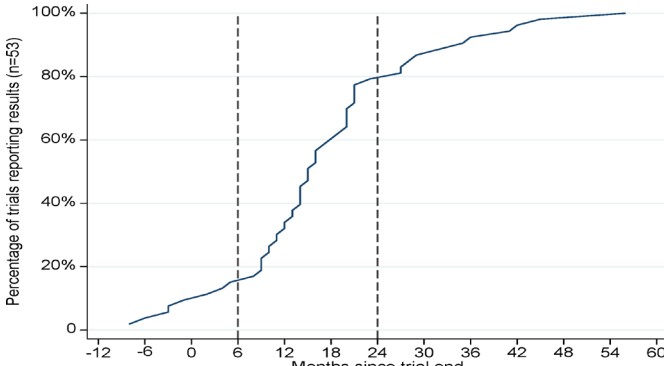

**Figure 3** Period in months between award/trial end and reporting results (n=53 completed* trials). * Nine completed trials that have not yet reported findings were excluded

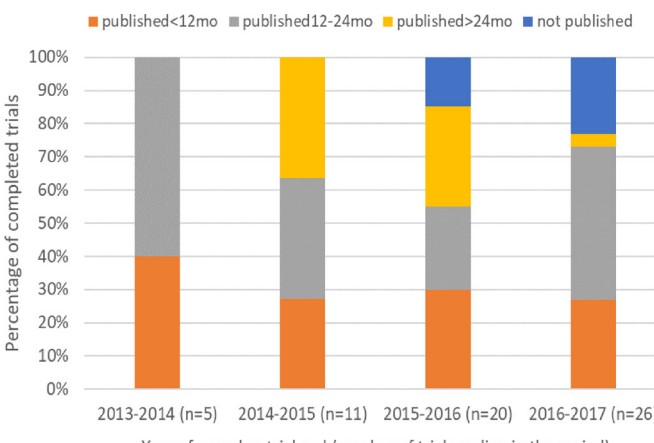

**Figure 4** Reporting (or publication) of main results by year when award ended.

the ISRCTN registry exclusively and to acknowledge MRC funding in the registry entry has yet to have a discernible impact.

### Public reporting of main trial results

Sixty-two awards/trials were completed before 01 February 2017, of which 42 (68%) publicly reported main results, or posted results in a registry, within 24 months of award/trial end. Only 18 (29% of 62) trials reported within 12 months of award/trial end. A further 11 (18% of 62) completed trials reported main results >24 months after award/trial end. Nine trials had not reported results (in March 2019).

Of 62 completed awards, 40 (65%) had a link to the main results or posted results in the registry (links=40; results posted=4; both were added for some trials).

The time taken to report results, for the 53 completed and reported trials, ranged from 8 months prior to 56 months after award/trial end, and the majority reported between 6 and 24 months after award/trial end (figure 3).

The PIs for the nine unreported trials were contacted for further information; seven reported that papers were progressing and publication was expected within the next 12 months. The period since trial completion ranged from 24 to 43 months; various reasons for the delay were highlighted, including the absence of a key researcher, negative findings or early trial termination. One award had included findings in a press release and symposium abstract but these were insufficiently detailed to be considered main results.

There was an increase in the percentage of trials reporting results before 24 months in 2016–2017 compared with most previous years; in contrast, the percentage of trials that reported within 12 months changed little over time (figure 4). In recent years, the percentage of unreported trials increased, reflecting the higher number of trials ending in these years as well as delays in reporting. Based on our experience between 2017 and 2018, it seems likely that the total number of unreported trials will reduce further with late publications.

## Data sharing

PIs reported creating 18 datasets for sharing with other researchers as an output from their clinical trial data, representing 10% of all 175 trials. Five of these datasets had been shared with researchers outside the original study team, one had been deposited in a repository for reuse by other researchers and one was available through a disease registry.

## DISCUSSION

As a signatory to AllTrials[12] and the WHO Joint Statement,[14] the MRC has made a commitment to the implementation of regular monitoring of researchers' compliance with the registration and reporting requirements for clinical trials set out in its *Policy on Open Research Data from Clinical Trials and Public Health Intervention Studies (2016)*.[15] Although the review found that all 175 trials funded by the MRC between 2011 and 2017 were registered, only half were registered in the WHO primary registry specifically named in the MRC award conditions and researchers often provided different information in the registry entry to that held by the funder. Summary results were reported within 2 years of completion for almost 70% of trials; most through academic papers but two-thirds also added links to these results in the trials registry entry. Despite this, the percentage of trials that had been completed for over 2 years but had failed to report was unacceptably high (15%). This underlines the need for renewed joint efforts by the MRC to address the barriers to the timely publication that were identified, which included delays in publishing after early trial termination or if findings were negative, and research staff absence or turnover.

## Registration

MRC-funded researchers are required to register trials in the ISRCTN registry because it permits registration of the wide range of study designs supported by the MRC and, as a WHO primary registry, contributes regularly to the ICTRP[11] maximising access to trials information. MRC policy explicitly states that researchers must add trial protocols and publications, as well as regularly reviewing and updating registry entries to ensure these remain accurate. An important positive finding was the decrease in the proportion of trials registering retrospectively, a trend that has also been noted by other authors and attributed to rising awareness among researchers of the need to register their trials.[23–25] However, our review is consistent with previous reports[26 27] that have highlighted inaccurate and incomplete registry entries as a significant cause for concern. This suggests that registries should review their guidance and controls around data entry by researchers to improve the consistency and accuracy of trial records. Disappointingly, a high proportion of trials in our review were not registered in the ISRCTN registry despite the MRC reimbursing the registration fee, and it was also noticeable that information in many registry entries was discrepant with that held by the MRC. Furthermore, only one-third of registry entries included a link to the trial protocol, despite early protocol publication being key to minimising publication bias and selective reporting of trial findings.[14 28] These findings suggest that researchers do not fully appreciate their responsibilities with regard to clinical trials transparency, specifically the need to regularly update registry entries so that they provide accurate and complete information to participants, the public and the research community. We also observed that the responsibility for complying with clinical trials transparency requirements falls largely on the shoulders of individual researchers and would therefore strongly endorse the call by Goldacre *et al*[29] for trial sponsors and research organisations which host trials to take greater responsibility for promoting and monitoring transparency, providing more administrative support, and raising awareness and compliance among researchers.

## Reporting results

Funder signatories to the WHO Joint Statement[14] committed to developing policies on timely publication, including an indicative timeframe of 24 months for publication in a journal and working towards a timeframe of 12 months for reporting summary results in a registry. Current MRC policy includes an expectation that the main trial results will be reported 'usually' within 12 months, but this is not mandated. European Union (EU) and US clinical trials legislation mandate the posting of trial results by 12 months after trial end[30 31] although this is often not achieved,[29 32] whereas other non-commercial funders have mandated time frames ranging from 12 to 24 months after trial completion.[7 33] Our review confirms 24 months as an achievable time frame for publication in peer-reviewed journals for MRC PIs; therefore, mandating a policy requirement for all trials to *either* publish *or* report findings in the registry by 24 months would encourage significant improvement on current practice. We also demonstrated that contacting researchers about reporting was an effective prompt that resulted in submission for publication. This is consistent with the findings of the national audit undertaken by the UK Health Research Authority (HRA)[25] and could be easily integrated into the routine annual review process by funders, registries and regulators. Although trials with null or negative findings often experience longer delays to publication,[34 35] there is increasing recognition by journals of the importance of null studies. This is underlined by the development of journals focusing solely on such studies, initiatives such as the recent special supplement of *Neurology* on the null hypothesis,[36] as well as the introduction of the 'registered reports' model to some journals.

## Data sharing

Funders are increasingly encouraging researchers to make their data available after study completion to maximise the opportunity for health improvement from public investment in research;[37] however, we found little

evidence of trial datasets being prepared or made available for reuse by researchers. In recent years, significant efforts have been made to develop guidance[38–41] and services to enable researchers to share data, including data repositories and online catalogues, independent access review committees, secure platforms for data access and analysis, and data anonymisation tools to reduce disclosure risk.[42–45] There have also been various initiatives to increase academic interest and innovation in data sharing, such as the SPRINT challenge.[46] However, as yet, these incentives to share data do not seem to be sufficiently persuasive and a central concern remains the failure to adequately recognise or 'credit' those who create important data resources for use by other researchers.[47 48]

### Sustainable and effective compliance monitoring

Conducting this review demonstrated that deriving a complete trial dataset for annual monitoring currently requires the cross-validation of data extracts from multiple existing sources. No single source contributed >85% of the final dataset, nor does any have sufficiently high levels of accuracy to be relied on as the sole data source for compliance monitoring at present. Importantly, PIs failed to report the registry number for their trial to the MRC for 71% of trials in this audit, indicating a concerning lack of compliance with this basic requirement.

Inaccuracy and incomplete data were often due to a failure by researchers to maintain and update registry entries as has previously been reported.[27 49] Although the ISRCTN registry does not provide a template for posting results, it was a reliable source of information about the publication of summary results as it actively followed-up with researchers to confirm the main results' publications for completed studies. In contrast, although both EUCTR and ClinicalTrials.gov provide templates for posting results directly into the registry, only four of the studies in our dataset (one in EUCTR and three in ClinicalTrials.gov) had completed these.

Although keyword searches successfully identified many trials, our review required a significant commitment of staff time to perform extensive internet searches, check the accuracy of data and cross-validate information from different sources. This included visual checks of registry entries and grant applications to ensure that studies were eligible trials and that information was complete; this was only possible with the help of Policy Interns undertaking placements as part of an MRC-funded PhD studentship. Importantly, over 30% of PIs had to be contacted directly one or more times to confirm that a study had progressed beyond the preclinical phase, or to provide registry numbers or publication details. Although initiatives using automated monitoring to identify trials which have failed to report findings have been pursued elsewhere,[29 49] it is not currently possible to 'automate' the compliance monitoring process for MRC awards. The challenge of introducing automated methods to monitor trial publications was also highlighted in the HRA national audit of ethical approvals.[25 50] Although it is not feasible for the

MRC to commit such a high level of staff and resources to carry out this manual verification process every year, the dataset created during the current review now reflects the complete clinical trials portfolio and can be supplemented each year with information about new funding awards. The key to further reducing the burden of compliance monitoring for funders and regulators is likely to be greater reliance on registry data; therefore, it is essential to focus on strengthening requirements for researchers to maintain accurate, complete and regularly updated registry entries for all ongoing trials.

### Strengths and limitations

As with the national audit undertaken by the UK HRA,[25] our review was based on a 'complete' denominator, comprising all MRC awards including a clinical trial during the review period, so accurately reflects registration and publication rates. PIs were contacted if there was any uncertainty about whether a study met the WHO definition of a clinical trial and studies that were not required to register under our policy were excluded, for example, non-interventional, genetic or biomarker studies or studies that failed to progress beyond the preclinical stages.

A limitation of our review dataset was the relatively small number of awards, in particular completed trials, during the period, which had an impact on analyses of publication rates and investigation of trends. We did not attempt to extend the dataset before 2011 as the MRC did not introduce applicant guidance on trial registration until 2013 so this would not have reflected contemporary practice; however, in future reviews, additional years of data will be added to the current dataset. We were also restricted by the information available, for example, we could only explore prospective registration in the ISRCTN registry as the other registries did not report this, however, we would expect the proportion prospectively registered to be similar across registries.

### CONCLUSIONS

As the current compliance review mainly included MRC awards before the publication of the new transparency policy, it is perhaps best considered as a reflection of existing practice. However, it will provide a clear benchmark for assessing improvement in clinical trials registration and reporting in future years. This review highlighted some key issues for funders consideration when establishing compliance monitoring including (1) developing the use of registry data as a primary source for compliance monitoring, (2) strengthening funder requirements for registry entries to be maintained, (3) improving identification of funded awards that include trials to ensure a complete denominator for evaluating compliance with registration, and (4) including the research organisation, registry number and award reference number in compliance monitoring reports to allow further details of individual trials to be found in public registries.

As a funder of a complex clinical trial portfolio, spanning early phase translational clinical trials as well as late phase trials involving behavioural, psychological and public health interventions, the MRC has a keen interest in promoting research transparency across all research involving human participants regardless of study design. An initial step has been to extend the MRC *Policy on Open Research Data from Clinical Trials and Public Health Intervention Studies* (2016) to include all clinical and public health interventional studies. We are now working towards the inclusion of explicit statements about transparency within the data management plans of all funded clinical and public health studies.

**Acknowledgements** The authors would like to thank staff at the ISRCTN registry and within the MRC Evaluation and International Teams for providing the data for the review.

**Contributors** RLK, RP and FR: conceived and designed the study. JM, KPH and RK: extracted and analysed data and all authors interpreted the data. RLK and RP: drafted the manuscript. All authors reviewed and approved the final manuscript.

**Funding** JM and KPH were funded by MRC Policy Internships. This research received no specific grant from any funding agency in the public, commercial or not-for-profit sectors.

**Competing interests** None declared.

**Patient consent for publication** Not required.

**Provenance and peer review** Not commissioned; externally peer reviewed.

**Data availability statement** Data are available upon reasonable request. The dataset is available on the MRC website as a downloadable file or on enquiry to the MRC.

**ORCID iD**
Rachel L Knowles http://orcid.org/0000-0002-5490-7682

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
