## [Reviewer comments · BMJ Open]

ARTICLE DETAILS

TITLE (PROVISIONAL)	Challenges for funders in monitoring compliance with policies on clinical trials registration and reporting: analysis of funding and registry data in the United Kingdom.
AUTHORS	Knowles, Rachel; Ha, Kam Pou; Mueller, Julia; Rawle, Frances; Parker, Rosa

VERSION 1 - REVIEW

REVIEWER	Kylie Hunter NHMRC Clinical Trials Centre, University of Sydney, Australia I am a staff member of the Australian New Zealand Clinical Trials Registry (ANZCTR)
REVIEW RETURNED	06-Nov-2019

GENERAL COMMENTS	Thank you for the opportunity to review this nicely written and interesting paper, which is an important contribution to global efforts to improve clinical trials transparency. I have only minor recommended revisions, as described below. Specific comments relating to Review Checklist: 6. Are the outcomes clearly defined? It is unclear which requirements you are analysing registration compliance against, e.g. MRC award conditions (2013), and/or MRC policy introduced in 2016, as the latter is more strict in terms of prospective registration. Refer to comment below relating to Page 6, line 50. Please also refer to comments relating to Page 6, line 55, and Figures 3 and 4, regarding how award/trial end/completion is defined. 9. Do the results address the research question or objective? If a key research question is to assess compliance with the MRC 2016 policy, then specific data (e.g. proportions) relating to prospective registration should also be included in the Results paragraph 'Trial registration'. (In addition to the point that there has been a reduction in trials registering retrospectively). General comments: Page 4, line 18 The reference to "ineffective or harmful drugs" here is too narrow. This sentence applies to all types of ineffective or harmful interventions that may be implemented (e.g. devices, procedures, etc.) not just drug treatments. Therefore, I suggest replacing
---

'ineffective or harmful drugs are released to market' with "ineffective or harmful interventions are implemented in practice' (or similar).

Page 4, line 20

I don't think that "reducing duplication of reporting" is a key role of trial registries (after all, duplicate reporting is likely better than no reporting!). Of far more importance is reducing selective outcome reporting, so this could be more appropriate to refer to here.

The 'C' in ICMJE stands for 'Committee' not 'Consortium'.

Page 4, line 46

It would be useful to know what proportion of trials in this initial review were 'prospectively' registered (not just registered) if this information is available, in alignment with the policy outlined in table 1.

Page 5-6 Data sources

This section is quite complex, but is nicely illustrated by the accompanying figure.

Page 5, line 59

If known, it would be useful to indicate the approximate proportion of registered UK trials that are registered on the registries searched, i.e. the ISRCTN registry, ClinicalTrials.gov and EUCTR. E.g. for registered Australian trials, about 95% are registered on ANZCTR or Clinical Trials.gov.

Also, did you consider searching WHO ICTRP which includes data from each of the registries listed plus all other primary registries? This would be a more comprehensive strategy.

Page 6, lines 37-39

It would be useful to expand on the methods used to ascertain reasons for trials not being identified by a particular source.

Page 6, line 50

In this section, you have indicated that you are analysing compliance with the MRC Policy on Open Research Data from Clinical Trials and Public Health Intervention Studies (2016). Table 1 indicates that the first requirement of this policy is "Prospective" registration, i.e. registration prior to recruitment of first participant. Therefore, if you are analysing compliance with this policy, objective 1 should refer to "Prospective" registration, not just registration in general.

In contrast, the 2013 MRC Award Conditions support "Retrospective" registration, and your audit covers the time period February 2011 and January 2017, so primarily prior to introduction of the 2016 policy.

Please be clear in this section about which requirements you are evaluating compliance against.

Page 6, line 55

It would be useful to define somewhere in your methods how trial/award end/completion is defined, given your results analyse publication of main results 12 and 24 months post this date, e.g. is this defined as when the funding period expires, when follow-up for

all outcomes is complete, etc.? This information would also aid interpretation of figures 3 and 4.

Page 6, line 59

Of importance to note here is that the 2013 MRC requirements supported retrospective registration of trials for treatments currently in use, for the sake of transparency, whereas the new 2016 policy requires prospective registration.

It would also be useful to know the month in 2016 when the new policy was implemented (e.g. Jan vs Dec), so we know how long it was applicable over the period of your dataset, which ends in 2017.

Page 8, line 30

The word 'category' is missing before '3' and 'categories 4' should be 'category 4'

It is unclear what the difference is between category 2 "a clinical study or trial" and category 5 "intervention study". Please clarify.

Page 8, line 43

This sentence suggests ISRCTN does not collect data on secondary funders? Were all funding source fields searched or only primary funding source?

Page 9, line 12

'acknowledged' should be 'acknowledge'

Page 9, 'Trial registration' section

It would be important and useful to mention here the proportion of the 175 trials that were registered prospectively, given this is the requirement stipulated in the MRC policy.

Page 10, line 8

"Based on previous evidence" - is there a reference for this?

Page 10, lines 34-35

"delays in publishing negative findings after early trial termination" In the results section, these are considered two separate barriers, i.e. negative findings OR early trial termination. Suggest clarifying in the discussion that these are separate.

Page 10, line 52

"a high proportion of trials in our review were not registered in the ISRCTN registry"

Did you consider whether this may be because there is a fee to register with ISRCTN, but not with other registries? Is the registration fee covered or waived for MRC funded studies?

Pages 11-12, lines 60-5

Does ISRCTN provide a template for posting results? Where were the four studies that used a template registered? This sentence would benefit from further clarification.

Page 13, line 8

Text beginning "we are now working..." appears to be a new sentence, so a full stop should be added and capitilisation of 'We'.

References

	A broad range of relevant references have been cited. 11 and 24 appear to be duplicates Figure 2 If possible, it would be useful to know the percentage of prospective vs retrospective registration for trials registered on other registries, not just ISRCTN. Figures 3 and 4 It is unclear whether the data is based on time when award ended or time when trial ended, as these terms are used interchangeably. In practice these dates could be quite different, so would be useful to clarify.
--	---

REVIEWER	Jake Checketts Oklahoma State University Center for Health Sciences, USA
REVIEW RETURNED	26-Nov-2019

GENERAL COMMENTS	Introduction: Very well written, easy to follow, and provides an excellent summary of the events in the UK leading to transparent prospective registration that somebody unfamiliar with this topic could understand and digest. Methods: While the methods are thorough and encompassing. I did find myself re-reading almost every paragraph to try and re-orient myself while reading. This is likely due to complex methodology as the authors used several methods to locate their final sample. Figure 1 clarifies the methodology nicely, and if the methodology could flow as well as Figure 1 it would be perfect. I am unsure if this is something the authors should fix due to the nature of the methodology, however if the authors' feel they could find a way for the methodology to read easier that would be appreciated by many readers I am sure. Results: The authors state that the trial registry number was only reported 29% of the time (line 38). To me, this is a big finding. If authors do not report their mandatory prospective trial registration number then what good is prospective trial registration? Reporting this number should be expected as it allows the reader to look up a trial and determine if they biased their results through outcome switching or other measures. Put simply: It does not matter if 100% of the studies were prospectively registered (as reported by the authors) if you cannot easily locate the trial registry number to look up the original prospective outcomes and measures. The first 2 sections of the results seem to be more narratively described than objectively reported in my opinion. Which again, is due to the complex methodology and different methods in which the authors found trials for inclusion. Discussion: I really enjoyed how the authors broke down each section of the discussion. The discussion (similar to the introduction) was easily read and broken up nicely. The intro and discussion are the "star" of this paper, and they were thoughtfully constructed and contain
---

	very relevant information that those unfamiliar with the subject could understand easily. Overall: This is an important topic, and the intro / discussion are well crafted, relevant, and easily understood. At times the methodology and results are hard to follow. If the authors could provide the same flow to their methods and results that they have in the intro and discussion the paper would have a wider reach as taking in these sections to full understanding really did take an excessive amount of time that I would have not given to a paper had I just been reading at my own leisure.
--	--

VERSION 1 – AUTHOR RESPONSE

Reviewer: 1

Specific comments relating to Review Checklist:

6. Are the outcomes clearly defined?

It is unclear which requirements you are analysing registration compliance against, e.g. MRC award conditions (2013), and/or MRC policy introduced in 2016, as the latter is more strict in terms of prospective registration. Refer to comment below relating to Page 6, line 50.

We analysed compliance against our most recent published policy (2016) and have now made this clear by adding 'with MRC's clinical transparency requirements as set out in the Policy on Open Research Data from Clinical Trials and Public Health Intervention Studies published in 2016 and to identify key areas for improvement' (page 4, lines 53-55). Although this policy was not in place when some of the trials in our dataset began, we were keen to understand how many existing trials were compliant with current policy and thus establish a benchmark for future audit.

Please also refer to comments relating to Page 6, line 55, and Figures 3 and 4, regarding how award/trial end/completion is defined.

We have addressed this below.

9. Do the results address the research question or objective?

If a key research question is to assess compliance with the MRC 2016 policy, then specific data (e.g. proportions) relating to prospective registration should also be included in the Results paragraph 'Trial registration'. (In addition to the point that there has been a reduction in trials registering retrospectively).

As we are assessing compliance against the MRC 2016 policy criteria, we have added information about the proportions of trials registered prospectively to the results on page 9, lines 36-39 as follows: 'The percentage of MRC trials registered prospectively in the ISRCTN registry increased over time: in 2011-2013 this was 22 (50.0%) of 44, in 2014-2015 it was 28 (63.6%) of 44 and in 2016-2017 it was 7 (87.5%) of 8 trials (Figure 2).' We could only report this for the ISRCTN registry as the other trials registries did not include a field for prospective/retrospective registration.

General comments:

Page 4, line 18

The reference to “ineffective or harmful drugs” here is too narrow. This sentence applies to all types of ineffective or harmful interventions that may be implemented (e.g. devices, procedures, etc.) not just drug treatments. Therefore, I suggest replacing 'ineffective or harmful drugs are released to market' with “ineffective or harmful interventions are implemented in practice’ (or similar).

Thank you for this suggestion. We agree that this is better phrasing and have amended the text at page 4, lines 18-19 accordingly.

Page 4, line 20

I don't think that "reducing duplication of reporting" is a key role of trial registries (after all, duplicate reporting is likely better than no reporting!). Of far more importance is reducing selective outcome reporting, so this could be more appropriate to refer to here.

Thank you for making this point. We have amended the text at page 4, line 20 to state: ‘selective outcome’.

The 'C' in ICMJE stands for 'Committee' not 'Consortium'.

We have corrected this in the text at page 4, line 21.

Page 4, line 46

It would be useful to know what proportion of trials in this initial review were 'prospectively' registered (not just registered) if this information is available, in alignment with the policy outlined in table 1.

As stated in our answer above, only the ISRCTN registry included a field stating whether a trial was retrospectively or prospectively registered. Neither ClinicalTrials.gov nor EUCTR reported this. Although we could have compared trial start dates and trial registration dates in the registry entry to make an assumption about whether a trial was prospectively registered, this would have relied on the accuracy of the reported registry dates and we could not validate this. Therefore we have reported the available information on retrospective and prospective registration for the ISRCTN registry only. We have added a statement clarifying this limitation in our discussion on page 12, line 56-59: ‘We were also restricted by the information available, for example we could only explore prospective registration in the ISRCTN registry as the other registries did not report this, however we would expect the proportion prospectively registered to be similar across registries.’

Page 5-6 Data sources

This section is quite complex, but is nicely illustrated by the accompanying figure.

Thank you. We recognise that this section is complex to understand and it is intended to be read alongside Figure 1 for clarification. We have also simplified the Data Sources section, following comments from Reviewer 2 also, and have added new subheadings and paragraphs in this section to

break up the text, as well as removing and simplifying text where possible. We have left sufficient detail so that our methods are transparent and reproducible.

Page 5, line 59

If known, it would be useful to indicate the approximate proportion of registered UK trials that are registered on the registries searched, i.e. the ISRCTN registry, ClinicalTrials.gov and EUCTR. E.g. for registered Australian trials, about 95% are registered on ANZCTR or Clinical Trials.gov.

Unfortunately we do not have this information for the UK. The MRC funds only a proportion of Phase 3 trials conducted in the UK.

Also, did you consider searching WHO ICTRP which includes data from each of the registries listed plus all other primary registries? This would be a more comprehensive strategy.

We did consider downloading data from WHO ICTRP but we are aware that many of our PIs were registered on ClinicalTrials.gov, which is not a WHO primary or partner registry. Therefore we elected to search trials registries individually to ensure that we did not miss any registrations and to enable us to collect some information for the audit that is not available in the ICTRP dataset. In future years, we will explore using ICTRP as an additional source or replacement for individual registry searches.

Page 6, lines 37-39

It would be useful to expand on the methods used to ascertain reasons for trials not being identified by a particular source.

We took a qualitative approach and have provided additional information to explain this (Page 6, lines 45-48): 'Information from this comparison, as well as reviewing how search parameters such as funder name were recorded, was explored for trials that were and were not identified by each source to determine the reasons why differences occurred.'

Page 6, line 50

In this section, you have indicated that you are analysing compliance with the MRC Policy on Open Research Data from Clinical Trials and Public Health Intervention Studies (2016). Table 1 indicates that the first requirement of this policy is "Prospective" registration, i.e. registration prior to recruitment of first participant. Therefore, if you are analysing compliance with this policy, objective 1 should refer to "Prospective" registration, not just registration in general.

This point is well-made and we have clarified our objectives (page 6, lines 55-56).

In contrast, the 2013 MRC Award Conditions support "Retrospective" registration, and your audit covers the time period February 2011 and January 2017, so primarily prior to introduction of the 2016 policy.

Please be clear in this section about which requirements you are evaluating compliance against.

We have clarified which policy we are auditing against at page 4 (lines 53-55) and page 6 (line 53). We audited compliance with prospective registration as stated in the MRC policy introduced in October 2016.

Page 6, line 55

It would be useful to define somewhere in your methods how trial/award end/completion is defined, given your results analyse publication of main results 12 and 24 months post this date, e.g. is this defined as when the funding period expires, when follow-up for all outcomes is complete, etc.? This information would also aid interpretation of figures 3 and 4.

We have now added a clearer explanation of trial completion/award end date to our methods at page 8, lines 23-29: 'The funding award end date was considered to be equivalent to the trial end date; however if the record in the registry stated that the trial end date was after the funding award end date, trials were considered completed on the trial end date. We refer throughout the paper to 'trial/award end' to capture this combined end point. Time to publication was therefore calculated using the award end date for most trials but using the trial end date recorded in the registry when this was later than the award end date.'

Page 6, line 59

Of importance to note here is that the 2013 MRC requirements supported retrospective registration of trials for treatments currently in use, for the sake of transparency, whereas the new 2016 policy requires prospective registration.

Thank you. As stated above, we have now clarified that we are auditing against the 2016 policy.

It would also be useful to know the month in 2016 when the new policy was implemented (e.g. Jan vs Dec), so we know how long it was applicable over the period of your dataset, which ends in 2017.

The policy was implemented in October 2016 so was in effect for only the final 4 months of the dataset. We have clarified this by adding 'October' on page 6, line 53. We consider this audit to be a benchmark for practice prior to the 2016 policy as stated in our discussion on page 13 (lines 5-9).

Page 8, line 30

The word 'category' is missing before '3' and 'categories 4' should be 'category 4'

We have corrected these errors.

It is unclear what the difference is between category 2 "a clinical study or trial" and category 5 "intervention study". Please clarify.

These studies were recorded in the MRC database as intervention studies or 'trial' grants, however were a mix of observational and economic substudies associated with trials/intervention studies, and community intervention studies. To explain category 5 better, we have added the following text at

page 8, lines 43-44: '(e.g. community interventions, economic evaluations of interventions and other non-randomised interventions)' and we have added 'intervention' at line 41 to explain category 2.

Page 8, line 43

This sentence suggests ISRCTN does not collect data on secondary funders? Were all funding source fields searched or only primary funding source?

ISRCTN collects information on secondary funders and all fields were searched. However the MRC participates in partnership funding (Joint Global Health Trials) where one funder is selected to manage the grant and PIs therefore named only the 'managing' funder in the registry entry. For accuracy, we have changed the sentence (page 8, line 54) to read: 'the MRC was not managing the grant'.

Page 9, line 12

'acknowledged' should be 'acknowledge'

This error has been corrected.

Page 9, 'Trial registration' section

It would be important and useful to mention here the proportion of the 175 trials that were registered prospectively, given this is the requirement stipulated in the MRC policy.

We have added the figures for prospective registration in the ISRCTN on page 9, lines 36-38. We cannot provide this figure easily for trials registered in ClinicalTrials.gov only as there is no data field reporting this. We now acknowledge this clearly in our limitations (page 12, lines 56-58) and suggest that we would expect similar proportions to be reported prospectively in ClinicalTrials.gov as in ISRCTN.

Page 10, line 8

"Based on previous evidence" - is there a reference for this?

We did not intend to refer to a publication and have changed the wording to now state (page 10, line 14): 'Based on our experience between 2017 and 2018, it appears likely..'

Page 10, lines 34-35

"delays in publishing negative findings after early trial termination"

In the results section, these are considered two separate barriers, i.e. negative findings OR early trial termination. Suggest clarifying in the discussion that these are separate.

We have amended the text at page 10, lines 39-40 to emphasise that these are separate barriers.

Page 10, line 52

"a high proportion of trials in our review were not registered in the ISRCTN registry"

Did you consider whether this may be because there is a fee to register with ISRCTN, but not with other registries? Is the registration fee covered or waived for MRC funded studies?

The fee is covered in full by the MRC therefore should not be a barrier. We have clarified how the fee is paid on page 10, line 57-58: 'despite the MRC reimbursing the registration fee'.

Pages 11-12, lines 60-5

Does ISRCTN provide a template for posting results? Where were the four studies that used a template registered? This sentence would benefit from further clarification.

ISRCTN does not provide a template but there is a facility for uploading a pdf with results. The 4 trials in our dataset that used a reporting template were registered and posted their results in EUCTR (n=1) and ClinicalTrials.gov (n=3). This information has been added on page 12, lines 10-16: 'Although the ISRCTN registry does not provide a template for posting results, it was a reliable source of information about the publication of summary results as it actively followed-up with researchers to confirm the main results publications for completed studies. In contrast, although both EUCTR and ClinicalTrials.gov provide templates for posting results directly into the registry, only four of the studies in our dataset (1 in EUCTR and 3 in ClinicalTrials.gov) had completed these.'

Page 13, line 8

Text beginning "we are now working..." appears to be a new sentence, so a full stop should be added and capitilisation of 'We'.

We have now corrected this (page 13, line 23).

References

A broad range of relevant references have been cited.

11 and 24 appear to be duplicates

Thank you for pointing this out. We have now corrected this.

Figure 2

If possible, it would be useful to know the percentage of prospective vs retrospective registration for trials registered on other registries, not just ISRCTN.

As stated earlier, this was not directly reported by other registries. We have therefore used the ISRCTN as a benchmark and assumed that the proportions are likely to be similar in other registries. This is now clarified in the text on page 8 (lines 8-10) and page 12 (lines 56-58).

Figures 3 and 4

It is unclear whether the data is based on time when award ended or time when trial ended, as these terms are used interchangeably. In practice these dates could be quite different, so would be useful to clarify.

We have clarified our definition of award end/trial end in the text on page 8, lines 23-29; this was a combined end point as we assumed the trial ended on the award end date unless the registry entry stated a later trial end date (indicating that the trial continued after the end of the award). In these cases, we did not use the award end date as the main results publications could not be published before the trial end.

Reviewer: 2

Introduction: Very well written, easy to follow, and provides an excellent summary of the events in the UK leading to transparent prospective registration that somebody unfamiliar with this topic could understand and digest.

Methods: While the methods are thorough and encompassing. I did find myself re-reading almost every paragraph to try and re-orient myself while reading. This is likely due to complex methodology as the authors used several methods to locate their final sample. Figure 1 clarifies the methodology nicely, and if the methodology could flow as well as Figure 1 it would be perfect. I am unsure if this is something the authors should fix due to the nature of the methodology, however if the authors feel they could find a way for the methodology to read easier that would be appreciated by many readers I am sure.

The text in the Methods (Data Sources) section on pages 5 and 6 is intended to be read alongside Figure 1. We have added new subheadings and paragraphs to the Data Sources section so that it aligns more closely with Figure 1. We have also removed and simplified the text where possible although we have left sufficient detail to ensure that our methods are transparent and reproducible.

Results: The authors state that the trial registry number was only reported 29% of the time (line 38). To me, this is a big finding. If authors do not report their mandatory prospective trial registration number then what good is prospective trial registration? Reporting this number should be expected as it allows the reader to look up a trial and determine if they biased their results through outcome switching or other measures. Put simply: It does not matter if 100% of the studies were prospectively registered (as reported by the authors) if you cannot easily locate the trial registry number to look up the original prospective outcomes and measures.

Thank you. We agree this finding has not been given sufficient emphasis and have added a sentence in the discussion to highlight it (page 12, lines 5-7).

The first 2 sections of the results seem to be more narratively described than objectively reported in my opinion. Which again, is due to the complex methodology and different methods in which the authors found trials for inclusion.

The analysis in these sections was mainly qualitative however we consider this to be a valuable contribution to improving understanding of, and addressing, problems with compliance. We have

added more information about the methods (page 6, lines 45-48), as described earlier in response to Reviewer 1.

Discussion:

I really enjoyed how the authors broke down each section of the discussion. The discussion (similar to the introduction) was easily read and broken up nicely. The intro and discussion are the "star" of this paper, and they were thoughtfully constructed and contain very relevant information that those unfamiliar with the subject could understand easily.

Overall: This is an important topic, and the intro / discussion are well crafted, relevant, and easily understood. At times the methodology and results are hard to follow.

If the authors could provide the same flow to their methods and results that they have in the intro and discussion the paper would have a wider reach as taking in these sections to full understanding really did take an excessive amount of time that I would have not given to a paper had I just been reading at my own leisure.

Thank you for these helpful comments which have guided us in our effort to improve the readability of the manuscript. We have revised the text in the section 'Data Sources' on pages 5-6 by adding subheadings and removing detail within the text. Figure 1 provides additional detail to supplement this text.

VERSION 2 – REVIEW

REVIEWER	Kylie Hunter NHMRC Clinical Trials Centre University of Sydney Australia I am a Senior Project Officer for the Australian New Zealand Clinical Trials Registry (ANZCTR)
REVIEW RETURNED	10-Dec-2019

GENERAL COMMENTS	Thank you for addressing these comments. Just a note for future: although ClinicalTrials.gov is not a Primary or Partner Registry in the WHO Registry Network, it is a WHO ICTRP data provider and provides updated datasets to the portal on a weekly basis.
--